# Trehalose Reduces Nerve Injury Induced Nociception in Mice but Negatively Affects Alertness

**DOI:** 10.3390/nu13092953

**Published:** 2021-08-25

**Authors:** Vanessa Kraft, Katja Schmitz, Annett Wilken-Schmitz, Gerd Geisslinger, Marco Sisignano, Irmgard Tegeder

**Affiliations:** 1Institute of Clinical Pharmacology, Faculty of Medicine, Goethe-University Frankfurt, 60590 Frankfurt, Germany; kraft@med.uni-frankfurt.de (V.K.); schmitz@med.uni-frankfurt.de (K.S.); wilken-schmitz@em.uni-frankfurt.de (A.W.-S.); geisslinger@em.uni-frankfurt.de (G.G.); marco.sisignano@med.uni-frankfurt.de (M.S.); 2Fraunhofer Institute for Translational Medicine and Pharmacology ITMP, 60596 Frankfurt, Germany; 3Fraunhofer Cluster of Excellence for Immune Mediated Diseases (CIMD), 60596 Frankfurt, Germany

**Keywords:** spared nerve injury, nociception, IntelliCage, learning and memory, fungal sugar

## Abstract

Trehalose, a sugar from fungi, mimics starvation due to a block of glucose transport and induces Transcription Factor EB- mediated autophagy, likely supported by the upregulation of progranulin. The pro-autophagy effects help to remove pathological proteins and thereby prevent neurodegenerative diseases such as Alzheimer’s disease. Enhancing autophagy also contributes to the resolution of neuropathic pain in mice. Therefore, we here assessed the effects of continuous trehalose administration via drinking water using the mouse Spared Nerve Injury model of neuropathic pain. Trehalose had no effect on drinking, feeding, voluntary wheel running, motor coordination, locomotion, and open field, elevated plus maze, and Barnes Maze behavior, showing that it was well tolerated. However, trehalose reduced nerve injury-evoked nociceptive mechanical and thermal hypersensitivity as compared to vehicle. Trehalose had no effect on calcium currents in primary somatosensory neurons, pointing to central mechanisms of the antinociceptive effects. In IntelliCages, trehalose-treated mice showed reduced activity, in particular, a low frequency of nosepokes, which was associated with a reduced proportion of correct trials and flat learning curves in place preference learning tasks. Mice failed to switch corner preferences and stuck to spontaneously preferred corners. The behavior in IntelliCages is suggestive of sedative effects as a “side effect” of a continuous protracted trehalose treatment, leading to impairment of learning flexibility. Hence, trehalose diet supplements might reduce chronic pain but likely at the expense of alertness.

## 1. Introduction

Trehalose is a disaccharide found in bacteria, yeast, insects, fungi, and plants but not in mammals [1,2]. Two glucose molecules are linked via an alpha-glycosidic linkage, providing heat and pH resistance [3]. The degrading enzyme trehalase cleaves trehalose and provides glucose as an energy resource in microorganisms such as mycobacteria but not in mammalians [1,3]. Nevertheless, trehalase was also found in the brain of mice [2] and in the gastrointestinal tract of mice and humans [4], likely originating in part from the gut microbiome [5,6,7]. The bioavailability of ingested trehalose is low [8,9] and may depend on bacteria colonization [10]. Nevertheless, trehalose-based diets in mice have positive cardiometabolic [4,11,12], neuroprotective [13], and anti-inflammatory properties [14].

Trehalose blocks the cellular glucose transport via the solute carrier protein 2A (SLC2A), thereby mimicking a starvation state [15]. Lowering glucose availability in cells leads to the activation of adenosine monophosphate-activated protein kinase (AMPK) and the phosphorylation of the autophagy-activating kinase ULK1, converging on the stimulation of autophagy [3,15] independently of mammalian target of rapamycin (mTOR) activation [16,17].

In addition, trehalose has gained popularity as a putative anti-dementia supplement owing to its functions as an inducer of Transcription Factor EB (TFEB)-mediated autophagy [15], a mechanism suggested to contribute to the removal of protein waste such as beta amyloid or protein aggregates [12,18,19,20,21]. Owing to these effects, trehalose was studied in a number of neurodegenerative mouse models [15,22] and was found to protect against cognitive decline in models of Alzheimer’s disease [23,24,25,26,27], prolong survival in models of motoneurons disease [17,28,29,30], and reduce brain damage in a model of traumatic brain injury [31,32]. TFEB is a transcription factor that induces lysosomal biogenesis [18], autophagosome formation, and autophagic flux [17] by the induction of a number of autophagy and lysosomal genes [33,34] including that coding for the neuroprotective protein progranulin [35], which assists in the autophagic flux as a cargo [36,37,38,39]. 

In previous studies, we showed that transgenic overexpression of progranulin in sensory neurons of the dorsal root ganglia reduces nerve injury-evoked neuropathic pain in mice and enhances the autophagic flux and axonal regrowth after sciatic nerve damage [39,40], suggesting that a trehalose-mediated enhancement of TFEB-dependent autophagy and progranulin upregulation may provide a safe, diet-based protection against neuropathic pain after nerve injury. Therefore, here, we studied the effects of trehalose treatment via drinking water in the mouse Spared Nerve Injury model of neuropathic pain in terms of nociception, motor functions, physiology, spatial and social behavior, activity, learning, and memory.

## 2. Materials and Methods

### 2.1. Mice and Trehalose Treatment

Experiments were performed in female C57Bl6/J mice (8-week-old at the start of the study) (Charles River, Sulzfeld, Germany). Female mice were used because IntelliCage experiments require housing of social groups (16 per IntelliCage) without social fights over several weeks. Mice were housed, 3–5 mice per cage, at constant room temperature (22 ± 2 °C), 45–65% humidity, and a 12 h circadian schedule with lights on from 7:00 a.m. to 7:00 p.m. Food and water were freely available. Trehalose was administered continuously via the drinking water at a concentration of 5%. Saccharose (5% in tap water) was used as a placebo control. Treatments started the day after surgery. Experiments were approved by the local Ethics Committee for Animal Research (Regierungspräsidium Darmstadt, Germany; V-54-19 c 20/12-FK1104, approval date 25 July 2018) and adhered to European regulations. The experiments followed the 3R ARRIVE principle and adhered to the guidelines of GV-SOLAS for animal welfare in science. The time courses of behavioral experiments are shown in Appendix A.

### 2.2. Spared Nerve Injury (SNI)

At the age of 8 weeks, mice were randomly assigned to trehalose and placebo groups and were subjected to a left-sided spared nerve injury (SNI) of the sciatic nerve. Surgery was done under 1.5–2% isoflurane anesthesia with local lidocaine anesthesia of the skin. The left sciatic nerve and its three peripheral branches were exposed. The tibial and common peroneal nerves were tightly ligated and distally sectioned. The sural nerve remained uninjured. Trehalose and placebo treatment were initiated after surgery.

### 2.3. Analysis of Nociception

Behavioral experiments were performed by an investigator who was blinded to group allocations. Mechanical and thermal nociception were assessed before Spared Nerve Injury (SNI) and after 3, 7, 14, 21, and 28 days after SNI and after 3 and 6 months (time courses in Appendix A). 

Mechanical nociception was assessed with a Dynamic Plantar Aesthesiometer (Ugo Basile, Gemonio VA, Italy). In this test, a filamentous rod is pushed against the plantar hind paw with a linear ascending force (0–5 g at 0.5 g/s) and is then maintained at 5 g until paw withdrawal or a cut-off set at 30 s. Three trials were averaged per time point. 

The sensitivity to painful heat stimuli was measured in the Hargreaves test (IITC Live Science, Woodland Hills, CA, USA) to assess the development of heat hyperalgesia. Mice were exposed to a radiant heat source which was placed underneath the plantar surface of the right or left hind paw. The paw withdrawal latency was measured 3 times with intervals >30 min, and the average was used for further analysis. The cut-off was set at 20 s.

A hot plate or cold plate (IITC Life Sciences) was used to further assess heat nociception and cold allodynia. The temperature of the surface was set to 52 °C for the hot plate test and at 0 °C for the cold plate test. The acetone test was used to measure cold allodynia as described [41]. The duration of the acetone response was monitored for 90 s with a stopwatch and started directly after the application of acetone. Licking, lifting, or shaking the paw were considered as acetone-evoked behavioral responses. 

### 2.4. Primary Neuron Culture

Primary neuron-enriched cultures of dorsal root ganglia (DRG) neurons were prepared by dissecting the DRGs of adult mice into Neurobasal medium (Gibco) with 5% penicillin/streptomycin (Sigma-Aldrich, Germany). DRGs were digested with 2.5 mg/mL collagenase A (Millipore), and 1 mg/mL dispase II (Invitrogen) before treatment with DNase (Sigma, 250 U per sample). Triturated cells were centrifuged through a 15% fat-free bovine serum albumin solution (Sigma), plated, and cultivated on poly-L-lysine-coated cover slips in serum-free Neurobasal medium containing 1x B27 supplement (Gibco), 1% penicillin/streptomycin (Sigma-Aldrich), 200 ng/mL nerve growth factor (Gibco), and 2 mM L-glutamine (Gibco) at 37 °C and 5% CO_2_. Neurons were subjected to calcium imaging 24 h after plating.

### 2.5. Calcium Imaging

Calcium influx in sensory neurons upon stimulation with capsaicin is a biological correlate of nociceptive sensitivity of peripheral sensory neurons. Capsaicin causes burning pain in vivo via activation of the transient receptor potential channel, family V member 1 (TRPV1).

Calcium fluxes were measured fluorometrically in cultured adult DRG neurons with Fluo-4. Fluo-4 is a dynamic single-wavelength fluorescent calcium indicator. The increase of fluorescence intensity at the emission wavelength of 525 nm, upon excitation at a wavelength of 480 nm, reflects a rise in cytoplasmic calcium. Calcium-imaging experiments were performed with a Leica calcium-imaging setup. Images were captured every 2 s. Cells were loaded with Fluo-4, incubated for 40 min at 37 °C, and washed three times with Ringer solution. Coverslips were then transferred to a perfusion chamber with a flow rate of 1–2 mL/min at room temperature. Baseline ratios were recorded for 200 s, before the application of 50 nM capsaicin (Sigma-Aldrich, Germany) to activate TRPV1 ion channels for 20 s. After wash-out with Ringer solution, cells were perfused with 50 mM KCl (high K^+^) to assess depolarization-evoked calcium currents and the viability of the neurons. Data are presented as fold changes in absorbance normalized to baseline ratios. The analysis encompassed 70 neurons per group of each 4 independent experiments. The maximum, the time of maximum, and the area of the fold increase versus the time curve were calculated with GraphPad Prism 9. The time courses and areas were used for statistical comparison.

### 2.6. Phenomaster Ananlysis of Voluntary Wheel Running, Feeding, and Drinking

We analyzed voluntary wheel running and drinking and feeding behavior in a Phenomaster^®^ cage (TSE systems, Bad Homburg, Germany) 7 weeks after SNI. Mice were adapted to the drinking bottles for 5 days in their home cage and to the Phenomaster^®^ cage for 24 h before the 24 h test period. The drinking volume and feeding were monitored with PC-controlled precision scales.

### 2.7. Motor Functions

Motor coordination was assessed with a constant-speed Rotarod (Ugo Basile), at 25 rpm and a cut-off latency of 180 s. Mice were adapted to the apparatus through 2–3 trainings runs the day before the test runs. The fall-off time was averaged from three tests. 

Motor functions were further assessed by the analysis of the travel paths in a Barnes Maze and a three-chamber maze used for testing spatial learning and sociability, respectively. 

A horizontal balance beam test was used to further assess motor skills. The rods had decreasing diameters of 35, 20, and 15 mm and were 60 cm in length. Mice were placed on the levitating end facing the end. They had to turn and walk back to a box. The turning and transit times were measured with a stopwatch. Motor functions were assessed 30–40 days after SNI.

### 2.8. Open Field Test

The Open Field Test (OFT) was done 7 weeks after SNI in a box of 50 × 50 × 38 cm according to the standard procedure. The area was divided in a center and a border, which were defined by the software. Mice were placed in the center at test start and were allowed to move freely for 10 min. The behavior was monitored with a video camera connected to a PC-based tracking software (VideoMot2, TSE, Bad Homburg, Germany). The time spent in the compartments, the entries, and the travel paths were analyzed. 

### 2.9. Elevated Plus Maze

Anxiety-like behavior was assessed 7 and 21 weeks after SNI in a standard Elevated Plus Maze test (EPM, TSE, Bad Homburg, Germany). The EPM consisted of two open and two closed arms extending from a central platform. At the trial start, a mouse was placed into the central area facing an open arm and was allowed to move freely for 10 min. Its behavior was monitored with VideoMot2 (TSE, Bad Homburg, Germany). Times and entries to the open and closed arms and the overall paths were analyzed.

### 2.10. Barnes Maze

Spatial learning and memory was assessed 10–11 weeks and 38 weeks after SNI in a standard Barnes maze (TSE, Bad Homburg, Germany) as described [42,43]. The protocol consisted in habituation, target learning, and reversal learning. During habituation, mice were placed in the middle of the maze in a transparent box for 20 s, were then guided to the target hole, and allowed to escape and stay in the shelter for 1 min. The habituation consisted in 6–8 trials, 2 per day. For target acquisition (1 trial per day for 4 days), mice were placed in the middle of the maze in a dark box for 20 s and were then allowed to explore the maze for 5 min (cut off) or until they entered the shelter. For the subsequent reversal learning period (1 trial per day for 4 days), the escape box was moved to the opposite side of the maze. We used the video tracking system for analysis VideoMot2 (TSE, Bad Homburg, Germany). Major parameters were time and the length of the travel paths until escape.

### 2.11. Sociability and Social Memory

Sociability was assessed 40 weeks after SNI as described [44]. The social test apparatus consisted of a rectangular, three-chamber box according to recommended specifications. The central chamber was separated from the side chambers by removable partitions with doors allowing the animal to move freely between chambers. The “stranger” mouse was presented within a grid enclosure that allowed interactions and nose contact but no attack or fight. The enclosures had an internal diameter of 7 cm, a height of 15 cm, and grey PVC bottom and roof. 

Mice were habituated to the test box for 3 days, each time for 10 min. For the habituation, the animal was placed in the middle chamber with the dividers closed to allow exploration of the middle chamber for 5 min. For the sociability test, an unfamiliar adult mouse (stranger 1) was placed inside the grid enclosure in one of the side chambers. An empty grid enclosure was placed in the opposite chamber. The dividers were then raised, and the test mouse was allowed to move freely among the chambers for 10 min. For testing of the preference of social novelty, the original stranger mouse (stranger 1) remained in its grid enclosure on one side of the apparatus, and a new unfamiliar mouse (stranger 2) was placed in the grid enclosure on the opposite side. The behavior was observed for 10 min. Stranger-1 and Stranger-2 were female mice like the test mice. The visits to the side chambers and time spent in each chamber with/without close contact with the stranger were recorded and analyzed with VideoMot2 (TSE, Bad Homburg, Germany). 

### 2.12. IntelliCage Tasks and Schedule

The IntelliCage experiments were started 26 weeks after SNI and lasted 2–3 months. The cages (NewBehavior AG, Zurich, Switzerland) consisted of four operant corners, each with two water bottles behind doors and light-emitting diodes (LEDs) above the doors. The doors controlled the access to the water bottles. The mice were tagged with radio-frequency identification (RFID) transponders which were red on corner entry. The software (IntelliCage Plus 2020, TSE, Bad Homburg, Germany) recorded corner visits, nosepokes, and lickings and executed user-defined pre-designed tasks. 

To avoid an interruption of trehalose or placebo treatments, mice were adapted for only 2 d with free access to every corner, with all doors open and tap water and food ad libitum (free adaptation, FA). Subsequently, mice were trained in an easy Place Preference Learning (PPL1) task, in which trehalose–water or placebo–water were presented in each two corners on both sides. The respective other doors remained closed. The assignment to the correct corners re-established the treatment of the respective group. After a learning period of 13 days, the correct corner was switched to the opposite side, and the difficulty was increased (PPL2). Access to the water bottle was now restricted to one corner. Because live monitoring revealed low activity and learning difficulties in the trehalose group, the experiment was interrupted for a home cage break. After 10 days, mice returned to the IntelliCage, which had not been cleaned, and started a PPL3 protocol, which was identical to PPL2 but with a rewarding corner opposite to that of PPL2. After 11 days of learning PPL3, mice were again assigned to a new correct corner (PPL4) and observed for further 7 days.

The number of visits and nosepokes were analyzed to assess overall activity, and the number of licks revealed the drinking behavior and the success rate. The percentage of correct nosepokes was analyzed per day to assess accuracy. In addition, a probability test was used to assess the number of trials that were needed to achieve the success criterion set to 10% above random success. Type 1 and type 2 errors were set to 0.05.

### 2.13. Statistics

Group data are presented as mean ± SD or mean ± sem, the latter for behavioral time courses. Summary data are presented as box plots and show the interquartile range, median, maximum, and minimum and are overlayed with the scatters of individual mice. The sample size was *n* = 16 per treatment group for nociceptive behavior, motor functions, OFT, and Barnes Maze. The experiments were done in two sequential experimental cohorts, each consisting of *n* = 8 per treatment group. For further, more complex tests (IntelliCage, EPM, Social, Phenomaster), the experimental cohorts were assigned to different tests, resulting in group sizes of *n* = 8 per treatment. Data were analyzed with Graphpad Prism 9.0, TSE Analyzer, and FlowR for IntelliCage experiments. Data distribution was assessed with the Shapiro–Wilk test, and data followed a normal distribution. Treatment groups were compared with unpaired, 2-sided Student’s *t*-tests. Time courses or multifactorial data were submitted to 2-way analysis of variance (ANOVA) using, e.g., the factors ‘time’ and ‘treatment’. Subsequently, groups were mutually compared using post hoc *t*-tests and adjustment of the *p*-value according to Šidák. For predefined comparisons (trehalose versus placebo), P was not adjusted for the between-subject factor, because only two groups were compared. Asterisks in figures show statistically significant differences between treatment groups. 

## 3. Results

### 3.1. Trehalose Long-Term Treatment Is Well Tolerated

General well-being, drug tolerance, and motor functions are prerequisites for the interpretation of nociceptive behavior and were ascertained throughout the observation time. Body weights were equal (Figure 1A), and there were no differences in drinking volumes (Figure 1B), feeding (Figure 1C), or voluntary wheel running (VWR) (Figure 1D). Trehalose water was marginally sweet and could not be differentiated from placebo water, which was equally marginally sweet. Blind taste tests had random results. Nevertheless, the 24 h drinking volume increased for both groups (8–9 mL/d). For comparison, tap water volumes range between 3 and 5 mL/d for C57BL6 mice of this age. The drinking volume was more variable in the trehalose group during the 24 h Phenomaster observation, but licking in IntelliCages was similar (please see below).

Voluntary wheel running showed similar engagement in rewarding running in both groups but suggested that trehalose-treated mice were somewhat less active (paired analysis of mice tested simultaneously). Therefore, the mice were subjected to further motor tests. The running time on a constant-speed Rotarod was equal in both groups (Figure 2A). There were also no differences in the turn-around time and transit time on balance beams, irrespective of the diameter (Figure 2B,C). In addition, the travel paths showing gross locomotion and daytime activity in Maze tests (OFT, EPM and 3-chamber Social) were equal (Figure 2D). Overall, body weights, physiology, and motor tests suggested that 5% trehalose was well tolerated and had no obvious side effects.

### 3.2. Trehalose Reduces Nociception in the SNI Model of Neuropathic Pain

Paw withdrawal latencies were equal at baseline before SNI and before the start of trehalose or placebo, and mice were randomly allocated to the treatment groups (Figure 3). After SNI, thermal and mechanical paw withdrawal latencies of the ipsilateral paw (Figure 3A,B, left) dropped in both groups and reached a minimum 2–3 weeks after injury. The decrease of the paw withdrawal latency (PWL) was stronger in the placebo group, indicating that SNI-evoked nociceptive hypersensitivity was reduced by trehalose. PWLs remained constant on the contralateral sides in both groups. The antinociceptive efficacy of trehalose also manifested in the hotplate test, which measures acute heat “pain” but is less sensitive for SNI-evoked hypersensitivity and does not differentiate between ipsi- and contralateral paws (Figure 3C). Responses to cold stimulation, tested in the cold plate test (Figure 3D) and in the acetone test, were similar in both groups (Figure 3E). The data suggest that trehalose treatment reduced SNI-evoked mechanical and heat hypersensitivity but not cold allodynia.

### 3.3. Trehalose Has No Effect on Calcium Fluxes in Primary Sensory Neurons

The positive therapeutic effects of trehalose are believed to be mediated through enhancement of TFEB-mediated autophagy. Pro-autophagy mechanisms also help to overcome pain persistence in the SNI model [39]. To assess the direct effects on nociceptive neurons, we used calcium imaging in primary DRG neurons. Pretreatment of the cultures with trehalose or placebo had no effect on the calcium influx evoked by capsaicin to stimulate TRPV1 channels (Figure 4A,B) or upon treatment with a high concentration of K^+^ to stimulate depolarization-mediated calcium currents (Figure 4A,C). The time courses of intracellular calcium ([Ca^2+^]i) were congruent. There were no differences of time to peak, maximum influx, or area under the peak curves. The lack of difference suggested that the antinociceptive effects of trehalose were mediated through cortical (central) inhibitory mechanisms. 

### 3.4. Trehalose Reduces Alertness and Accuracy in IntelliCage Learning Tasks

We used IntelliCages to assess putative central inhibitory mechanisms that manifest at the behavioral level in a home cage and social environment (Figure 5). Mice were subjected to place preference learning (PPL) tasks. Treatments were maintained throughout the observation period. To assess dark-phase and daytime activity, visits, nosepokes, and licking were analyzed in 12 h bins matching the 12 h light/dark cycle. The circadian rhythms were disrupted in the first PPL1 phase in both groups but reinstated in the subsequent PPL tasks. Further interruptions occurred upon reversal of the rewarding corner. It was obvious from the beginning that trehalose-treated mice made fewer nosepokes than vehicle-treated mice (Figure 5A, middle). This lower explorative behavior became stronger and was reinstated after a home cage break. Lower activity was also revealed in less visits, mostly in the post-break period. Trehalose-treated mice had very strong spontaneous preferences and did not switch to other corners when required by the PPL tasks. As a result, learning behavior (proportion of correct trials) was substantially reduced in the easy PPL-1 task when two corners were rewarded and in the final PPL-4 task, when a corner, which had not been rewarded in any previous period, was rewarded (Figure 5B). 

The behavior showed overall lower activity of trehalose-treated mice, which was associated with low learning flexibility and lower accuracy. Learning difficulties were also revealed in the flat learning curves (Figure 6A), the mean accuracy per module/task (Figure 6B), and the higher number of trials that were needed to reach the criterion of success, which was set at 10% above random. The differences between the groups were stronger but not restricted to the dark phase.

### 3.5. Trehalose Reduces Social Interest

Standard maze tests were used to further assess the effects of trehalose on behavior (Figure 7). The tests were done during daytime. There were no differences between the treatment groups in OFT (Figure 7A), EPM (Figure 7B), and Barnes Maze (Figure 7C). In the first part of the sociability test, trehalose-treated mice spent more time in the empty compartment and less in the social compartment as compared to placebo-treated mice (Figure 7D, left). There were no differences between the treatment groups in the second part of the social test (Figure 7D, right). Mice of both groups preferred the novel social partner over the familiar one. Overall, the maze tests suggest that the behavioral effects of trehalose are more evident in a social context (sociability tests and IntelliCages) and are predominantly revealed in the dark activity phase. Observations in IntelliCages are more precise than those in maze tests.

## 4. Discussion

We show in the present study that a protracted continuous trehalose administration in drinking water to mice reduced nociceptive hypersensitivity after sciatic nerve injury but negatively affected overall activity and learning performance in IntelliCages, suggesting a negative impact of trehalose on alertness or vigilance. The antinociceptive effects of trehalose were not mediated through direct inhibitory effects on sensory neurons as assessed by calcium imaging, suggesting central inhibitory effects on cortical nociceptive circuits, which would agree with a lowering of cortical activity and behavioral slowness. 

The cognitive differences between the treatment groups were mainly manifested in IntelliCage observations but not in standard maze tests, except for the sociability test. IntelliCages have the advantage of allowing the observation of social groups of mice in home cage environments continuously over weeks during dark phase and light phase, whereas Maze tests allow observing a single mouse in a non-familiar environment for 10 min during daytime. The results showed the superiority of continuous PC-based measurements of behavior over video-based maze methods. It may be regarded as a disadvantage that so far only female mice were studied in IntelliCages because male mice engage in social fights. 

The lowering of nosepoking activity in IntelliCage was not caused by motor dysfunctions as revealed by rotarod, balance beam, and travel paths in maze tests and would not affect nociceptive paw withdrawal latencies which are driven by reflexes. 

For treatment of neuropathic pain, one would like to avoid sedation, but so far, the available drugs for neuropathic pain, including opioids and pregabalin or other anti-epileptic drugs, are associated with sedation. Non-steroidal anti-inflammatory drugs which are free of central side effects do not reduce neuropathic pain [45]. Unfortunately, there are no studies of opioids or pregabalin in IntelliCages because the technique is newer than these old drugs. While dampening of activity is unwanted for a pain killer, the sedative effects of trehalose may be advantageous for diseases associated with pathological hyperactivity, such as neurodegenerative diseases [42,43,46]. The reduction of hyperactivity might have contributed to the previously observed positive effects of trehalose in behavioral studies in mouse models of dementia such as Alzheimer’s disease [25,26] or Huntington’s disease [28]. 

It is of note that SNI, per se, does not affect behavior in IntelliCages in place preference learning and reversal learning tasks [47], and the placebo-treated mice behaved normally for their age in comparison with previous studies [47,48]. Hence, although the drinking volume was increased for both groups in Phenomaster cages owing to the mildly sweet taste of the placebo and trehalose water, 24 h lickings in IntelliCages were within the normal range for the mice age [47,48]. There was no evidence that the rewarding appeal of trehalose water was lower than that of placebo water. 

Mechanistically, the majority of studies addressing trehalose as a putative neuroprotective drug focused on trehalose-mediated enhancement of autophagy to remove or reduce the burden of pathological proteins and aggregates such as amyloid beta and Huntingtin [13,17,18,26,30,49,50,51,52]. A recent study showed that trehalose imposed a low-grade lysosomal stress, resulting in the activation of TFEB-dependent lysosome biogenesis [21]. The authors suggest a direct effect of trehalose, which was detected as lysosomal cargo and mildly increased the luminal pH [21]. 

Positively, an increase of lysosomal biogenesis may help injured neuron to degrade protein waste by increasing the autophagolysosomal flux and capacity. The degradation processes are protracted in somatosensory neurons, owing to their long axons. We have shown previously that progranulin-mediated autophagy helps to overcome the persistence of neuropathic pain-like behavior in mice [39]. Autophagosomes are constantly formed at the tip of neurons and mature along their travel path towards the soma [53,54]. It may be speculated that trehalose assists in this flux by increasing TFEB [19,35]. However, the time. and site-specific functions of TFEB in neuropathic pain are currently unknown. 

It has to be considered that raising lysosomal pH interferes with lysosomal enzyme activity and degradation of its cargoes [55]. If it occurs alternatingly, the negative effects may be overridden by increasing the number of lysosomes, as suggested by Jeong et al. [21]. Our treatment schedule was continuous in the drinking water, which caused per se day/night fluctuations of the dose but, nevertheless, was constant. In addition to direct cargo effects, trehalose was also shown to mimic starvation by blocking glucose transporters [15,56]. Intermittent fasting is a popular approach to stimulate autophagy in the brain and thereby reduce the accumulation of pathological proteins [57,58,59]. Intermittent fasting has also been suggested to support the current pain treatments [60]. A continuous trehalose administration over an extended period may mimic long-term nutrient deficiency for neurons, and one may speculate that neurons will try to save energy and reduce their activity. We assume that antinociceptive effects and the in vivo sedation-like behavior reflect sedation-like effects on excitatory neurons, based on previous studies using in vivo electrophysiologic recordings in behaving mice [61,62,63]. However, we did not directly study cortical activity under trehalose. Based on our behavioral results and on mechanistic studies of trehalose-mediated TFEB enhancement, it appears that the positive effects of trehalose may be associated with a drop of vigilance if it is administered continuously over weeks. However, it remains to be proven in future studies that an intermittent administration of trehalose still provides antinociception after nerve injury while avoiding a drop of alertness and cognitive performance.

## 5. Conclusions

We conclude that trehalose-containing drinking water is a well-tolerated treatment that reduces neuropathic pain in mice but is accompanied by a mild loss of alertness.

## Figures and Tables

**Figure 1 nutrients-13-02953-f001:**
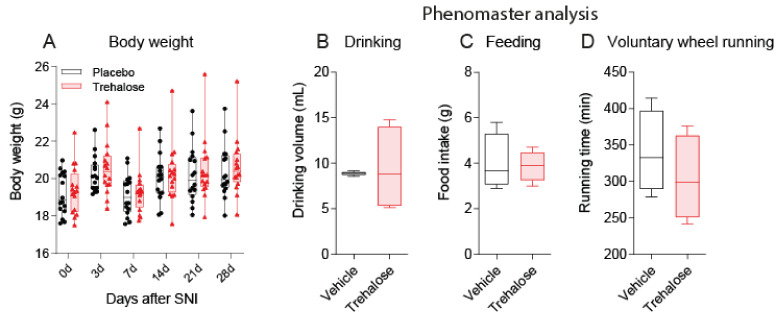
Body weights and Phenomaster analysis. (**A**): time course of body weights after Spared Nerve Injury (SNI). Two-way ANOVA showed no differences between the treatment groups. The scatters show individual mice (*n* = 16 per group). (**B**): Phenomaster analysis of the 24 h drinking volume. (**C**): Phenomaster analysis (7 weeks after SNI) of 24 h feeding. (**D**): Phenomaster analysis of voluntary wheel running (VWR). The box shows the interquartile range, the line the median, and the whiskers the 10th–90th quantile, (*n* = 8 per group). Data were compared using unpaired, 2-tailed Student’s *t*-tests.

**Figure 2 nutrients-13-02953-f002:**
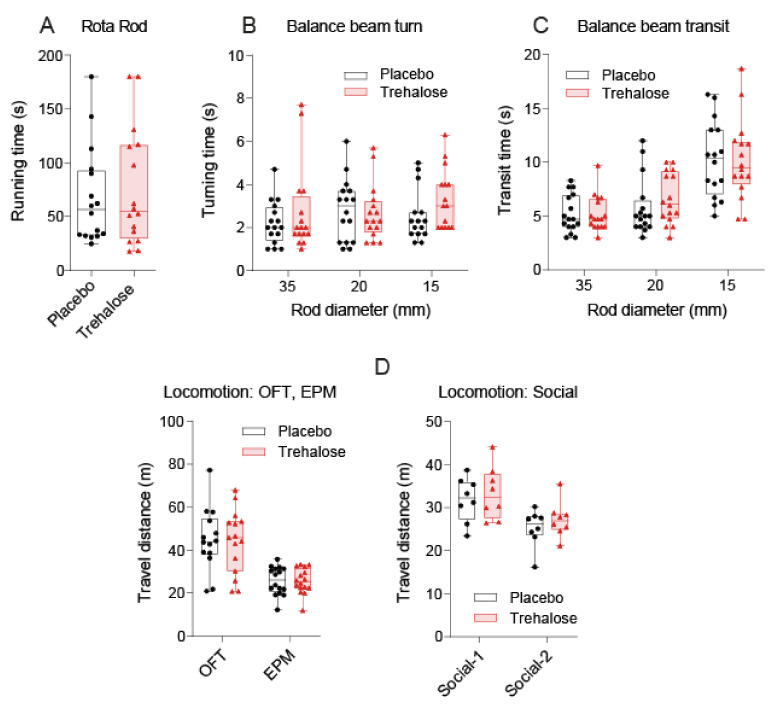
Motor behavior in trehalose. and placebo-treated mice 5–6 weeks after SNI. (**A**): Running time on a constant speed Rotarod. The fall-off latency did not differ between groups. (**B,C**): Turn around and transit time on a static balance beam with decreasing diameters (35, 20, 15 mm). (**D**): Travel distances in the Open Field Test (OFT), Elevated Plus Maze (EPM) tests, and 3-Chamber test of sociability and social memory. The box shows the interquartile range, the line the median, the whiskers minimum to maximum, the scatters individual mice (*n* = 16 per group, *n* = 8 per group for the social test). Data were compared with unpaired, 2-tailed Student’s *t*-tests or 2-way ANOVA for balance beam tests, using “diameter” X “treatment” as within- and between-subject factors. There were no differences between the treatment groups.

**Figure 3 nutrients-13-02953-f003:**
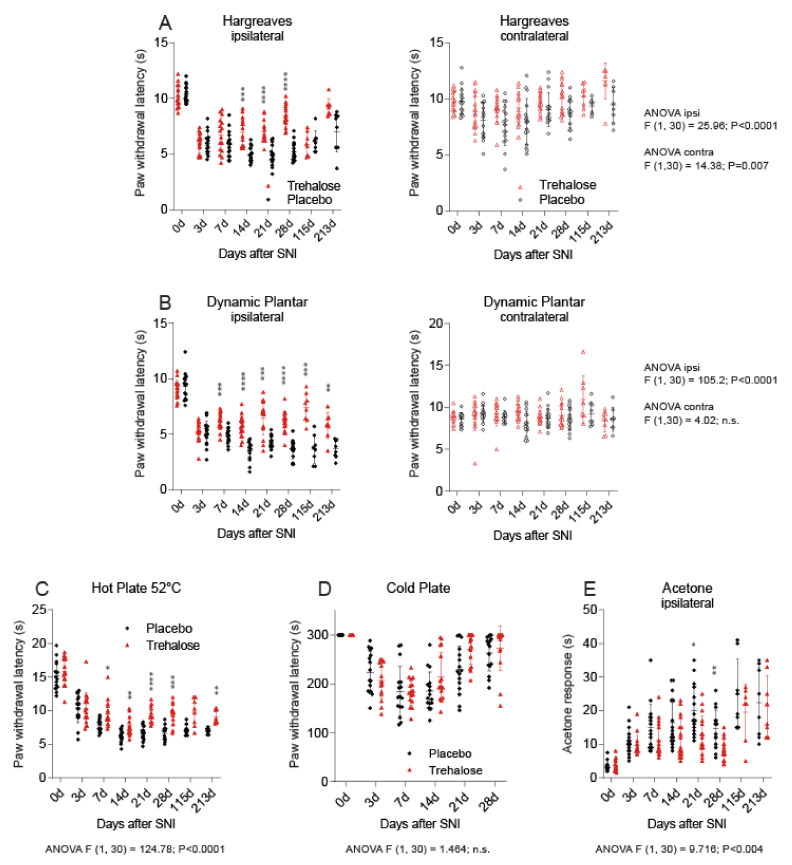
Nociceptive behavior of trehalose- and placebo-treated mice using the SNI model. Mice were treated continuously with 5% trehalose or placebo via drinking water, starting after surgery. Paw withdrawal latencies were assessed at baseline and up to 6 months after SNI. The scatter plots show individual mice (*n* = 16 per group), the thick line is the mean. (**A**): The Hargreaves test shows paw withdrawal latencies of the ipsilateral (left) and contralateral (right) hind paw before and after SNI upon stimulation with radiant heat. (**B**): Paw withdrawal latencies upon mechanical stimulation in the Dynamic Plantar test. (**C**): Hotplate latencies at 52 °C of the hotplate surface. (**D**,**E**): Cold pain hypersensitivity using a cold plate at 0 °C and in the acetone test. Data were compared with 2-way ANOVA for repeated measurements, using “time” X “treatment” as within- and between-subject factors, and subsequent post hoc analysis with P adjustment according to Šidák. Asterisks indicate statistically significant differences between the treatment groups. * *p* < 0.05, ** *p* < 0.01, *** *p* < 0.001., **** *p* < 0.0001, n.s. not significant.

**Figure 4 nutrients-13-02953-f004:**
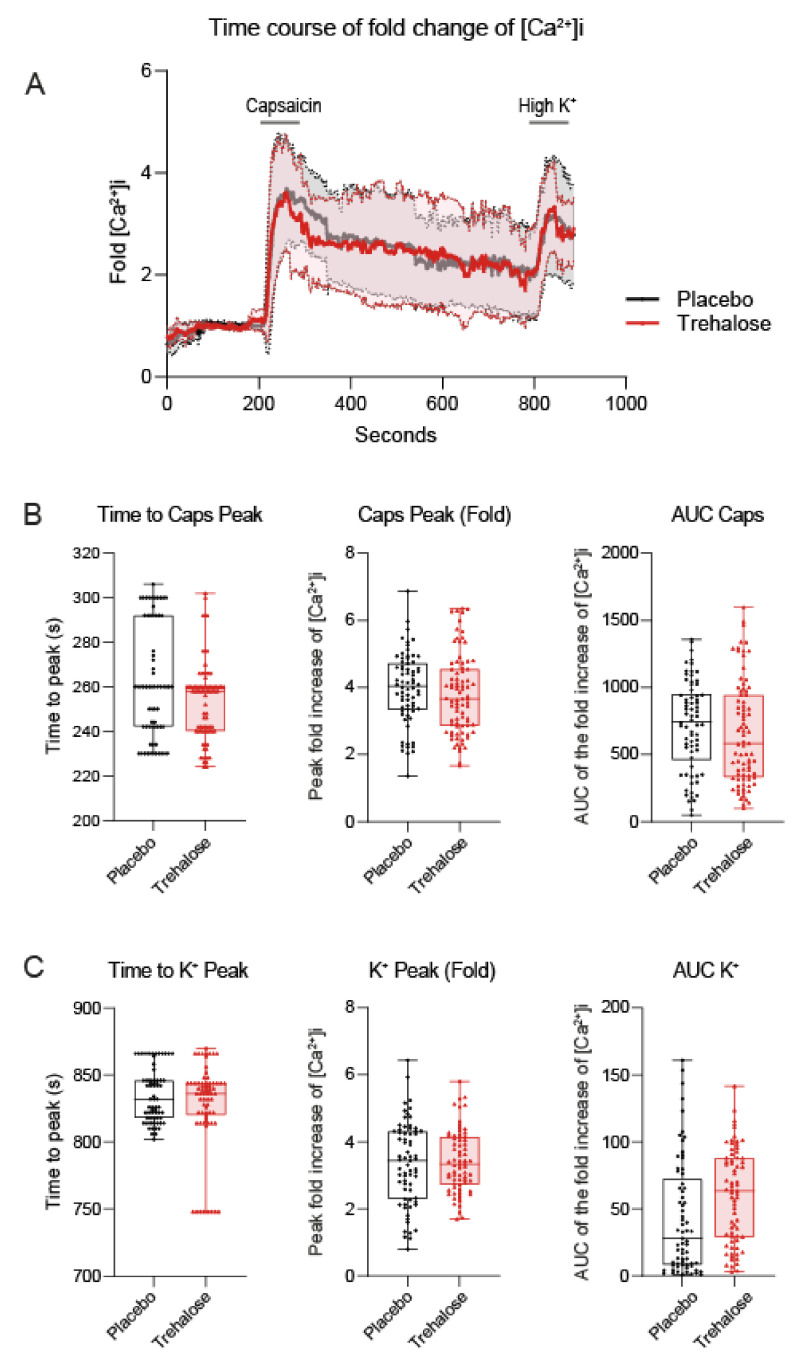
Calcium fluxes upon capsaicin (Caps) stimulation in primary neurons treated with trehalose or placebo. Neurons were prepared from the dorsal root ganglia (DRG) of naïve mice and were treated for 24 h with 10 mM trehalose or placebo before the analysis of calcium fluxes. (**A**): Time course of the fold change of intracellular calcium [Ca^2+^]i, which shows the change of Fluo-4 absorbance versus baseline. Baseline was measured for 200 s, followed by 20 s capsaicin (Caps) to assess TRPV1-mediated calcium influx, and finally highly 50 mM potassium (K^+^) for 20 s to assess depolarization-mediated calcium influx. The thick line is the mean, the filled areas show the SD. (**B**,**C**) Time to peak after capsaicin or highly concentrated K^+^, maximum change of [Ca^2+^]i, and the area under the curve (AUC) of capsaicin or highly concentrated K^+^ peaks. Data of *n* = 70 neurons per group were compared with unpaired, 2-tailed Student’s *t*-test. The box is the interquartile range, the line the median, the whiskers minimum to maximum, the scatters show individual neurons.

**Figure 5 nutrients-13-02953-f005:**
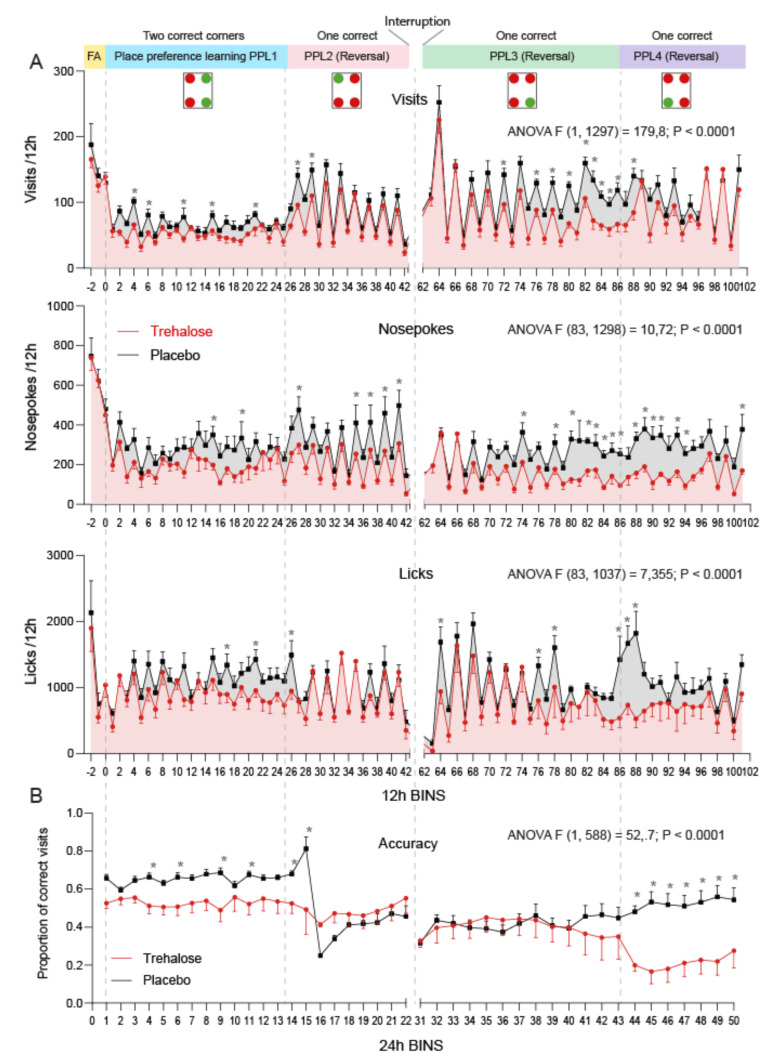
Activity and accuracy in Place Preference Learning (PPL) tasks in IntelliCages. Experiments started 26 weeks after SNI. (**A**): Time course of Visits, Nosepokes, and Lickings per 12 h intervals of trehalose- and placebo-treated mice (*n* = 8 per group). The fluctuations show the circadian rhythms. After a short free adaptation (FA), mice were subjected to place preference tasks as depicted in the upper part of the scheme. The drawings show exemplary corner assignments: red for the forbidden corner, green for the rewarding corner(s). Trehalose and placebo were administered via drinking water in rewarding bottles. During FA, all mice received tap water. The observations in IntelliCages were interrupted for 10 days in PPL2/3 to observe reinstatement of the behavior after a home cage interval. (**B**): Time course of the proportion of correct visits in PPL tasks as depicted in A. Data are the means and sem and were compared with 2-way ANOVA for “time” x “treatment” and subsequent post hoc analysis for trehalose versus vehicle. Asterisks show *p*-values < 0.05.

**Figure 6 nutrients-13-02953-f006:**
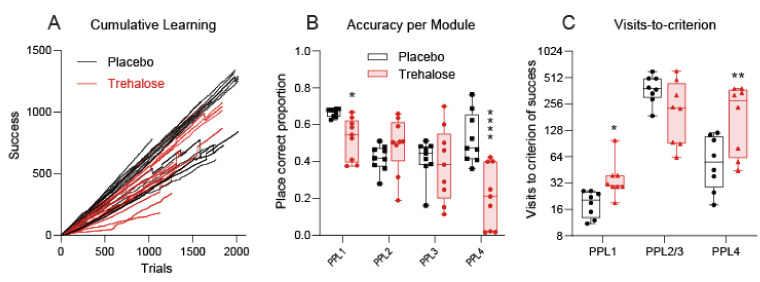
Learning in IntelliCage Place Preference Learning (PPL) tasks. (**A**): The XY plots shows the cumulative number of trials versus the cumulative number of successes, analyzed for each PPL period. Hence, each mouse is represented by four lines. The steepness of the learning curves is lower for trehalose-treated mice, particularly in PPL-1 and PPL-4. (**B**): Accuracy (proportion of correct visits) per learning module. (**C**): Number of trials needed to reach the criterion of success which was defined as 10% above random, i.e., 60% in PPL-1 and 35% in PPL-2, 3, and 4. PPL2/3 were analyzed together. The box shows the interquartile range, the line the median, the whiskers minimum to maximum, the scatters individual mice. Data in B and C were compared with 2-way ANOVA for “module” x “treatment” and subsequent post hoc analysis for trehalose versus vehicle, using an adjustment of alpha according to Šidák. Asterisks show multiplicity-adjusted *p*-values, * < 0.05; ** < 0.01, **** < 0.0001.

**Figure 7 nutrients-13-02953-f007:**
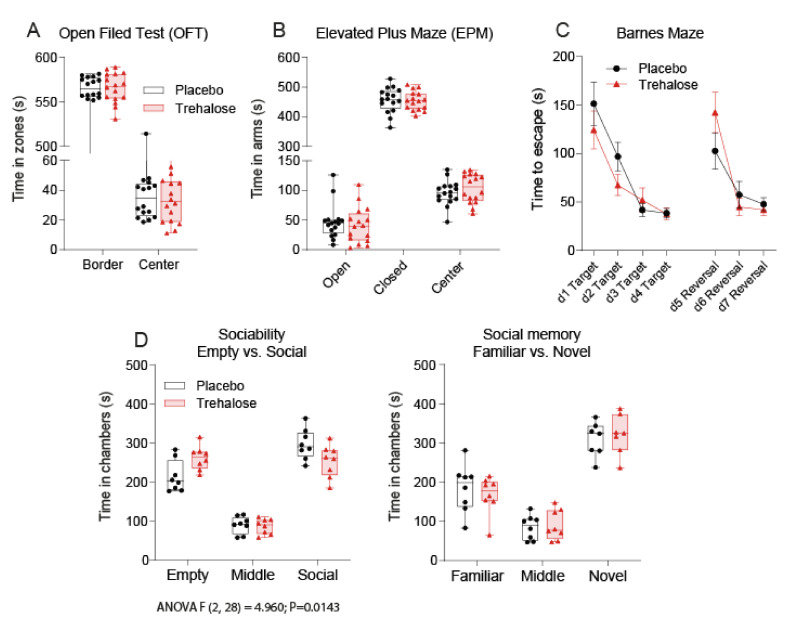
Behavior in Open Field, Elevated Plus Maze, Barnes Maze, and sociability and social memory tests. (**A**): Times spent in the center and border zones in OFT (7 weeks after SNI). (**B**): Times spent in the open and closed arms and in the center area in EPM test (7 weeks after SNI). (**C**): Time needed to escape in the shelter box in a Barnes Maze (8–11 weeks after SNI) in four learning trials (d1–d4) and three reversal learning trials (d5–d7). Data are means and SD. (**D**): Time spent in the compartments in a three-chamber maze test of sociability and social memory (40 weeks after SNI). The first part (Sociability) shows the preference of a social partner over an empty compartment. The second part (Social Memory) shows the preference of a novel social partner over the familiar mouse. The box is the interquartile range, the line the median, the whiskers minimum to maximum, the scatters show individual mice (*n* = 8 per group). OFT and EPM were done twice, hence each mouse is represented by two dots. Data were compared with 2-way ANOVA. OFT, EPM and Barnes were did not differ between groups.

## Data Availability

Data are presented in the paper. Raw data shall be made available on reasonable request. No omics datasets were generated or analyzed during the current study.

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
