# Peer review of "Trehalose Reduces Nerve Injury Induced Nociception in Mice but Negatively Affects Alertness"

_nutrients, 2021, doi:10.3390/nu13092953_

Round 1

Reviewer 1 Report

The manuscript deals with the effects of continuous trehalose administration via the drinking water in the Spared Nerve Injury model of neuropathic pain in mice. The findings show that continuous trehalose administration with the drinking water reduces nociceptive hypersensitivity after sciatic nerve injury but it is accompanied by negatively overall activity and learning performance suggesting a negative impact on alertness or vigilance. The authors hypothesis is that intermittent administration of trehalose could avoid a drop of cognitive performance.

My concern regards the use of a single concentration of Trehalose (5%) and female mice alone. Furthermore, the sample size (n = 8 per treatment group) can affect the reliability of the results.

My suggestion is that the work needs to be completed by studying a concentration range of Trehalose and comparing male and female behavior, at least for the more significant assays.

Author Response

Reviewer#1

The manuscript deals with the effects of continuous trehalose administration via the drinking water in the Spared Nerve Injury model of neuropathic pain in mice. The findings show that continuous trehalose administration with the drinking water reduces nociceptive hypersensitivity after sciatic nerve injury but it is accompanied by negatively overall activity and learning performance suggesting a negative impact on alertness or vigilance. The authors hypothesis is that intermittent administration of trehalose could avoid a drop of cognitive performance.

My concern regards the use of a single concentration of Trehalose (5%) and female mice alone. Furthermore, the sample size (n = 8 per treatment group) can affect the reliability of the results.

My suggestion is that the work needs to be completed by studying a concentration range of Trehalose and comparing male and female behavior, at least for the more significant assays.

Thank you very much for evaluation of our manuscript.

We have used n = 16 animals per group for nociceptive studies, motor studies, OFT and Barnes maze. For further behavioral experiments, cohorts had to be split. Therefore, n=8 were used for IntelliCage, Phenomaster and Social test. We have now included a Suppl. Excel file with Suppl. Tables showing the biography of experiments for each mouse of cohort-1 and cohort-2 and have now specified in the Methods under Statistics the sample sizes per experiment.

We agree that dose responses might be needed if it comes to clinical studies of trehalose in humans. However, we have not the Ethical allowance to use as many animals as would be needed for proper dose response studies in males and females. For 3 doses (each placebo versus one dose of trehalose) for two sexes one would need 12 x 16 animals. We can observe only two groups in the IntelliCage at a time, which is a 3-4 months lasting observation.  

It is not allowed to study male mice in IntelliCages because 16 mice are housed together in one cage, and male mice have strong social hierarchies and start fighting. There is no evidence from previous studies that effects of trehalose depend on animals' sexes.

We have added an explanation why female mice were used in the present study (in Methods under animals).

Reviewer 2 Report

The authors conducted a study on the pain sensitivity of trehalose based on previous studies of over expression of progranulin. Although it lacks the expected results, it is meaningful for pain-related studies of trehalose. 

This study sought to understand and show how changes in physiological behavior changed after thehalose, but did not describe the following points.

  1. In the SNI model, it is thought that the behavioral change appeared differently according to the developmental process of pain, but the results do not show information on when the behavioral test was performed.
  2.  In the comparison between groups, the effect of trehalose can be compared more clearly only if there is a comparison of the results of thehalose intake in animals using sham injury.
  3. It is argued that the increase in autphagy caused by TFEB is related to the pain reduction effect of threhalose, but inferences and experiments are not described.
  4. An explanation of the pain-reducing effect and side effects of trehalose, as well as a detailed explanation of the cause should be discussed in the discussion part.

Author Response

Reviewer#2

The authors conducted a study on the pain sensitivity of trehalose based on previous studies of over expression of progranulin. Although it lacks the expected results, it is meaningful for pain-related studies of trehalose.

This study sought to understand and show how changes in physiological behavior changed after thehalose, but did not describe the following points.

Thank you for the evaluation of our manuscript and your suggestions.

Trehalose reduced neuropathic pain-like behavior. Hence, the study does not entirely "lack the expected results", but shows side effects of trehalose.

In the SNI model, it is thought that the behavioral change appeared differently according to the developmental process of pain, but the results do not show information on when the behavioral test was performed.

The time course of nociceptive behavior is shown in Figure 3 in the x-axes labeling (Days after SNI). For the other experiments we have now added the information in the respective figure legend and in the Methods.  In addition, we include a mouse biography for cohort-1 (2 x 8 animals) and cohort-2 (2 x 8 animals) as a supplementary Excel file which shows the detailed time course of experiments for each individual mouse.

In the comparison between groups, the effect of trehalose can be compared more clearly only if there is a comparison of the results of thehalose intake in animals using sham injury.

Indeed, we have not done a "trehalose-only" (or trehalose-sham) group in this study because we can observe only two groups in the IntelliCage at a time and therefore we had to focus on the SNI-trehalose versus SNI-vehicle groups in this study.

It is argued that the increase in autphagy caused by TFEB is related to the pain reduction effect of threhalose, but inferences and experiments are not described.

We have assessed TFEB immunostainings in the DRGs (please see an exemplary image included for your information for review purposes - in the zip file). The tissue was obtained at the end of the behavioral studies and therefore the immunofluorescence studies reflect a single late time point. The number of neurons with TFEB immunoreactivity was higher in the trehalose group at this time. One may interpret this finding as higher lysosomal biogenesis, positively to reduce the burden of protein waste in injured neurons. However, we feel that the study of where and when TFEB upregulation occurs (in which cells at which site) and what it means for pain needs to be addressed in a separate study that is focused on TFEP. Presently, the function of TFEB in pain is unknown. Time courses in parallel with LC3b and p62 for autophagy and ATF3 as marker of axonal injury would be needed for interpretation. Here, we focused on the behavioral outcome and assessed tissue only at the end of the observations. Considering the transient nature of autophagy flux we think that our histology results do not fit into the present manuscript and we have therefore not included immunofluorescent studies.

An explanation of the pain-reducing effect and side effects of trehalose, as well as a detailed explanation of the cause should be discussed in the discussion part.

We have increased the discussion of putative mechanisms of trehalose mediated antinociceptive effects in the discussion as suggested. Presently, the mechanisms are unknown and hypotheses are based on previous studies which were done in cell culture systems. These studies revealed upregulation of progranulin and increase of TFEB mediated autophagy as the key results. Further studies show that trehalose triggers AMPK signaling. The mechanistic implications of trehalose for TFEB and AMPK are laid out in the Introduction to explain the motivation of our study.

Reviewer 3 Report

This study by Vanessa Kraft at al. is well written and clearly describes the activity of Trehalose on reducing nerve injury induced nociception in mice. I have just a suggestion regarding the size of the figures that are too small and is not easy to read in hard copy. Should be better to uniform all figures to figure 4, that in my opinion has the right size.

Author Response

Reviewer#3

This study by Vanessa Kraft at al. is well written and clearly describes the activity of Trehalose on reducing nerve injury induced nociception in mice. I have just a suggestion regarding the size of the figures that are too small and is not easy to read in hard copy. Should be better to uniform all figures to figure 4, that in my opinion has the right size.

We thank you for the evaluation of our manuscript and the positive feedback.

We have increased the size of all figures as suggested as far as it was possible within Nutrients' WORD template, which has to be used for the journal.

Round 2

Reviewer 1 Report

The authors state that "we have not the Ethical allowance to use as many animals as would be needed for proper dose response studies in males and females": I have not competence how to overcome this issue. My concerns continue to be. I acannot recommend the manuscript to publication